# Factors Related to Obstructive Sleep Apnea According to Age: A Descriptive Study

**DOI:** 10.3390/healthcare11233049

**Published:** 2023-11-27

**Authors:** Myoungjin Kwon, Jiyoung Kim, Sun Ae Kim

**Affiliations:** 1Department of Nursing, Daejeon University, Daejeon 34520, Republic of Korea; mjkwon@dju.kr; 2Biomedical Research Institute, Konyang University Hospital, Daejeon 35365, Republic of Korea; jykim270371@gmail.com; 3Department of Nursing, Korea National University of Transportation, Chungbuk 27909, Republic of Korea

**Keywords:** obstructive sleep apnea, middle aged, elderly, public health

## Abstract

This descriptive study aimed to identify the factors influencing obstructive sleep apnea (OSA) by age between middle-aged and elderly people. These groups have not been evaluated separately until now. This study is a secondary analysis of data from the Eighth Korea National Health and Nutrition Examination Survey 2021. Of the 3942 participants with OSA in this study, 2397 were middle-aged and 1545 were elderly, and 2509 had low risk and 1433 had moderate–high risk. Age-specific factors related to their OSA were identified using complex sample logistic regression. Factors associated with OSA in middle-aged individuals included the number of household members, smoking, subjective health, and subjective body image. Smoking and subjective health were significantly related to OSA in elderly people. Not smoking was 0.23 times less likely than smoking to be associated with OSA, and 0.3 times less likely when participants were healthy than when unhealthy. Hence, influencing factors differed depending on the age of individuals with OSA. Therefore, to develop public health measures, it may be more effective to establish intervention strategies to improve symptoms and prevent complications in middle-aged and elderly patients with OSA by distinguishing and applying influential age-specific factors.

## 1. Introduction

Obstructive sleep apnea (OSA) is a common recurrent breathing disorder that occurs during sleep, characterized by the collapse of the upper airway and cessation of airflow during sleep [1]. It has been reported that one billion middle-aged people suffer from OSA worldwide and that age is an important factor, with older people having more severe symptoms than younger people [2,3]. Particularly, 425 million people require treatment, which can result in economic and social burdens on individuals and society [2], posing a threat to public health. The prevalence of OSA has been reported to differ by age, with an increase in those aged <60 years but a decrease in those aged >60 years, prompting the need to investigate differences by life cycle [4]. Therefore, OSA management in middle-aged individuals may be important for health management and behavioral change. In previous studies, health behaviors, such as aerobic physical activity, smoking, and drinking, were reported to be related to the occurrence and severity of OSA, highlighting the importance of healthy lifestyle habits [5]. Physical activity has been reported to lower OSA risk. Studies have shown that increasing physical activity to 15 MET-h per week reduces OSA risk by approximately 10% in both men and women [5]. In addition, unhealthy eating habits centered on meat have been associated with moderate to severe OSA [6]. Hence, dietary behavior is a risk factor for OSA, and efforts at the public health and individual levels are needed to promote and maintain a healthy diet.

OSA is reportedly more common in men, those with obesity, and those with hypertension [7]. It has been reported that 26% of adults suffer from mild to severe OSA, and this prevalence is expected to increase further [8]. OSA causes ventilatory instability, intermittent hypoxia, and a rapid increase in blood pressure during sleep [9]. It is a risk factor for chronic diseases, such as diabetes, ischemic coronary artery disease, stroke, cancer, and hypertension [10]. OSA is a serious disease in itself, as well as due to its association with secondary cardiopulmonary diseases from the rapid rise in blood pressure. It is associated with serious heart-related sequelae, such as stroke and myocardial infarction [11]. Smoking is a factor that adversely affects health and is a major cause of cardiovascular diseases. Both smoking and OSA cause oxidative stress and endothelial dysfunction [12]. Therefore, it is natural that when these two factors combine, the negative effects become greater, and comorbidities increase. Smoking is associated with cardiovascular diseases. This unhealthy behavior can be corrected through individual efforts and environmental support, and it must be addressed in patients with OSA. Particularly, alcoholism and smoking are reportedly associated with high-risk OSA.

Smoking must be considered a disease-causing factor when trying to identify the factors influencing OSA. Additionally, OSA has been reported to be associated with low socioeconomic status and job loss [13], suggesting that economic and occupational statuses should be treated as important variables. In a previous study that identified psychological problems in patients with OSA, 20.5% of patients had suicidal ideation [14]; hence, it is important to confirm the relationship between OSA and suicidal ideation to prevent suicide. In addition to the emotional and psychological instability due to sleep problems, OSA has been associated with depression. A previous study has reported that the presence of OSA in patients with depression influences the treatment of depression and the effectiveness of medication [15].

Although age is a risk factor for OSA, research [16] has shown that OSA may be less severe in elderly people. This suggests the need for further research on the differences between middle-aged and elderly people regarding OSA. Additionally, it is essential to assess the risk of OSA and identify differences. A convenient and reliable tool is necessary to evaluate the risk of OSA. As a tool for assessing the risk of OSA, STOP-Bang comprises eight questions, providing a convenient means of evaluation [17]. Furthermore, a recent meta-analysis study [18] has validated its effectiveness, confirming its reliability as an assessment tool. Therefore, this study aimed to identify factors influencing OSA among middle-aged and elderly people, as they are major groups that, to the best of our knowledge, have not been evaluated separately until now. By confirming age-related factors, it may be possible to provide basic data for establishing age-appropriate OSA management and improvement strategies.

## 2. Materials and Methods

### 2.1. Study Design and Participants

This descriptive study aimed to identify factors associated with OSA by age. It is a secondary analysis of data from the Eighth Korea National Health and Nutrition Examination Survey (KNHANES VIII-3) 2021, a nationwide health survey conducted by the Korea Disease Control and Prevention Agency.

A trained investigator met with the participants in person and administered the survey using a self-report method.

As shown in Figure 1 below in total, 7090 people were included in the KNHANES VIII-3 (2021). Of these, 3942 participated in this study: 2397 middle-aged (40–64 years old) and 1545 elderly (65 years or older) individuals.

### 2.2. Study Variables

Risk of OSA was measured using the STOP-Bang tool, which comprises eight items [19]: snoring, fatigue, observed apnea, hypertension, body mass index (BMI; >35 kg/m^2^), age (>50 years), neck circumference (>41 cm), and sex (male). If the participants answered “yes” to three or more questions, they were classified into the “moderate–high-risk group”, and if they answered “yes” to less than three questions, they were classified into the “low-risk group”.

General characteristics included household economic level (upper, middle, or lower), education level (middle school or less, high school graduate, or college or higher), living with spouse, number of household members (1, 2, ≥3), occupation type (professional or office worker; service or sales worker; technical, agricultural, or simple labor worker; or unemployed), work type (day, night, or shift), working hours per week, weight change (loss, gain, or no change), binge drinking frequency (none, about 1/month, about 1/week, or almost every day), and smoking status.

Physical and psychological characteristics included diabetes, myocardial infarction, stroke, aerobic physical activity, daily sitting time, subjective health (healthy, normal, or unhealthy), subjective body image (thin, normal, or obese), stress (feeling less or feeling a lot), depression, suicidal thoughts, and anxiety. Aerobic physical activity was defined as moderate physical activity for at least 2.50 h per week, high-intensity physical activity for at least 1.25 h per week, or a mixture of moderate- and high-intensity physical activities (1 min of high-intensity work = 2 min of medium-intensity work) [20]. Anxiety was assessed using the Generalized Anxiety Disorder-7 (GAD-7) scale developed by Spitzer et al. [21]. The GAD-7 is a 4-point Likert scale with 7 items that measure the frequency of symptoms over the past 2 weeks. Higher scores indicated more severe anxiety.

### 2.3. Ethical Considerations

The National Health and Nutrition Examination Survey is a statutory survey on the health behaviors of people, the prevalence of chronic diseases, and the actual state of food and nutritional intake, conducted in accordance with Article 16 of the National Health Promotion Act. The 2021 National Health and Nutrition Examination Survey was conducted with the approval of the Research Ethics Review Committee of the Korea Disease Control and Prevention Agency (approval number: 2018-01-03-5C-A).

The Korea Disease Control and Prevention Agency provides only de-identified data; hence, individuals cannot be identified from survey data, in compliance with the Personal Information Protection Act and Statistics Act. The research data were downloaded from the National Health and Nutrition Examination Survey website with permission from the Korea Disease Control and Prevention Agency.

### 2.4. Statistical Analyses

As the National Health and Nutrition Examination Survey used a complex sample design method, a complex sample analysis that reflected strata, clusters, and weights was used to increase the accuracy of the estimation for data analysis. Data were analyzed using SPSS version 25 (IBM Corp., Armonk, NY, USA), and statistical significance was set at a *p*-value < 0.05. All analyses were performed in accordance with the recommendations of the Korea Disease Control and Prevention Agency. The differences and degrees of variables according to the participants’ age and degree of OSA were analyzed using complex sample descriptive analysis and the chi-squared test. Factors related to OSA according to age were identified using complex sample logistic regression.

## 3. Results

### 3.1. Comparison of Factors Related to OSA According to Age and OSA Risk Level

Among 2397 middle-aged people at risk of OSA, 690 were at moderate-to-high risk and 1707 were at low risk. Among 1545 older adults at risk of OSA, 743 were at moderate-to-high risk and 802 were at low risk.

There were significant differences in snoring, fatigue, observed apnea, hypertension, age, BMI, neck circumference, and sex according to the degree of OSA (Table 1).

### 3.2. Comparison of General Characteristics by Age and OSA Risk Level

A significant difference in education level was observed only among middle-aged individuals, and a significant difference in cohabiting status was observed only among the elderly. However, there were significant differences in the number of household members, job type, working hours per week, frequency of binge drinking, and smoking status between the groups.

In the middle-aged group, more participants with moderate–high risk of OSA had less than middle school education, compared with those with low risk. In the elderly group, the proportion of those who lived with a spouse was higher in the moderate–high-risk group than in the low-risk group. In both age groups, more individuals with a moderate–high probability of developing OSA had two household members, long working hours, and high rates of smoking and binge drinking, compared with those with a low probability of developing OSA (Table 2).

### 3.3. Comparison of Physical and Psychological Factors According to Age and OSA Risk Level

Sitting time per day, subjective health, subjective body image, stress, and suicidal ideation showed significant differences only among middle-aged participants.

Participants at moderate–high risk for OSA spent more time sitting, were more likely to be unhealthy and obese, were more stressed, and had more suicidal ideation, compared with those at low risk.

Moreover, there were significantly more participants with diabetes, myocardial infarction, and stroke among individuals with a moderate–high risk than among those with a low risk of OSA, in both the middle-aged and elderly groups (Table 3).

### 3.4. Factors Associated with Risk of OSA by Age

Factors associated with risk of OSA in middle-aged individuals included the number of household members, smoking, subjective health, and subjective body image.

For households with two members, compared with those with three, the probability of OSA was 1.57 times higher (95% confidence interval (CI): 1.09–2.27), and the probability of not smoking was 0.3 times lower than that of smoking (95% CI: 0.27–0.69). Moreover, the probability of OSA was 0.60 times lower for those who were healthy than for those who were unhealthy (95% CI: 0.39–0.90). Subjective body image was 0.55 times less likely to be normal than obese (95% CI: 0.40–0.76).

Smoking and subjective health were significantly related to OSA in elderly individuals. Compared with smoking, not smoking was 0.23 times (95% CI: 0.12–0.44) less likely to be related to OSA. In addition, being healthy was 0.3 times (95% CI: 0.12–0.78) less likely than being unhealthy to be associated with OSA (Table 4).

## 4. Discussion

OSA is a common sleep disorder characterized by partial or repeated upper airway obstruction during sleep, resulting in altered breathing patterns and intermittent hypoxia. To develop effective prevention and management strategies, it is essential to identify and understand the factors associated with OSA in different age groups. In this study, we investigated factors related to OSA in middle-aged and elderly individuals using KNHANES VIII-3 (2021) data. Weighted complex sample analysis was performed by assigning weights to correct bias and generalization errors.

Among middle-aged individuals, this study showed that the higher the level of education above high school, the lower the risk of OSA. The middle-aged group with an education level of middle school and below had a high risk of OSA. Although it was difficult to find literature showing that the educational level directly affects OSA, this relationship could be predicted through various mediating factors and relationships. Education level is closely related to socioeconomic status, lifestyle, and cognitive function; therefore, it can affect accessibility to medical services, such as diagnosis and treatment for OSA [22]. Additionally, lifestyle factors, such as diet, exercise, and stress management, may affect the risk of developing OSA and increase the awareness of the importance of maintaining a lifestyle that reduces the risk of obesity [23,24]. However, as there was no significant difference according to the education level in the elderly population, further in-depth research is needed. Unlike the middle-aged group, the highest risk in the elderly group was found among those living with a spouse. This suggests that living with a spouse helps to recognize OSA symptoms, and one is more likely to visit a hospital for diagnosis [25].

Working hours were associated with the severity of OSA in both middle-aged and elderly participants in the present study, consistent with the findings of a study that evaluated KNHANES 2020 data using the STOP-Bang score, that long working hours or working overtime increases OSA risk [26]. Additionally, it has been reported that occupations that require minimal physical activity due to sitting for long periods may contribute to OSA [27,28]. Nevertheless, OSA may be influenced by mediating factors other than age and working hours; therefore, further investigation is needed.

Smoking and alcohol consumption also affect OSA severity. Previous studies have also shown that smoking and alcohol consumption are closely related to OSA [23,29]. Since smoking and alcohol consumption are related to unhealthy lifestyles and the results of this study indicate that consuming more than one drink per month is associated with a high risk of OSA, healthcare interventions related to smoking cessation and alcohol abstinence among middle-aged and older adults should be strengthened.

Physical factors, such as diabetes, myocardial infarction, and stroke, were significantly related to OSA severity in both middle-aged and elderly individuals. This is consistent with the findings of other studies [23,30,31] and suggests that middle-aged and elderly individuals with these conditions should be screened for OSA.

Psychological factors, such as subjective health, subjective body image, stress, and suicidal thoughts, have been shown to influence OSA severity in middle-aged people. Specifically, self-perceived unhealthiness and obesity have been associated with a higher risk of OSA. A related study examining polysomnographic changes associated with weight loss in patients with severe OSA found that body image significantly improved with weight loss [32]. In addition, unlike the elderly, middle-aged people, who are at high risk of OSA, often have stressful and suicidal thoughts. These findings are consistent with those of previous studies [33,34,35]. However, these studies did not analyze middle-aged and older adults separately; hence, further research is required.

Anxiety had no effect on OSA severity in the middle-aged or elderly group. This finding differs from that of a study that found that OSA is closely related to anxiety, and that anxiety symptoms may interfere with the treatment of OSA [36,37]. In this study, anxiety was introduced for the first time in KNHANES VIII-3 (2021), which surveyed participants living in the community, not patients diagnosed and treated in hospitals. Therefore, it is expected that the mean value of anxiety will be low, and the significance level will be high. If more data on anxiety and OSA are accumulated in the future, more accurate and meaningful research results may be obtained.

This study was a secondary analysis study using KNHANES VIII-3 (2021) data and has the following significance. Age, reported as an important factor in OSA, was divided into middle-aged and elderly groups to identify age-specific factors, which can be used to develop a life-cycle-specific education program that considers age and establishes more accurate and detailed treatment and management plans for OSA. Second, through preventive management policies for physical and psychological factors that affect OSA, the quality of life of the public can be enhanced by improving sleep quality, increasing the diagnosis rate, and applying early interventions.

This study is limited in that it is a secondary analysis of KNHANES VIII-3 (2021) data, making it difficult to establish causality. However, we conducted the study using a large sample size, and the results can be generalized to a large population. In the future, longitudinal studies are needed to infer the causality of factors associated with OSA and enhance the strength of evidence.

## 5. Conclusions

The results of this study showed that the factors associated with OSA differed by age. The influencing factors analyzed by age may serve as basic data for age-based intervention strategies, as they reflect differences by life stage. In addition, variables identified as having the same impact or not may be useful for a step-by-step intervention strategy. Therefore, as a public health strategy, interventions should aim to improve symptoms and prevent complications in middle-aged and elderly individuals with OSA by distinguishing and applying age-specific factors.

## Figures and Tables

**Figure 1 healthcare-11-03049-f001:**
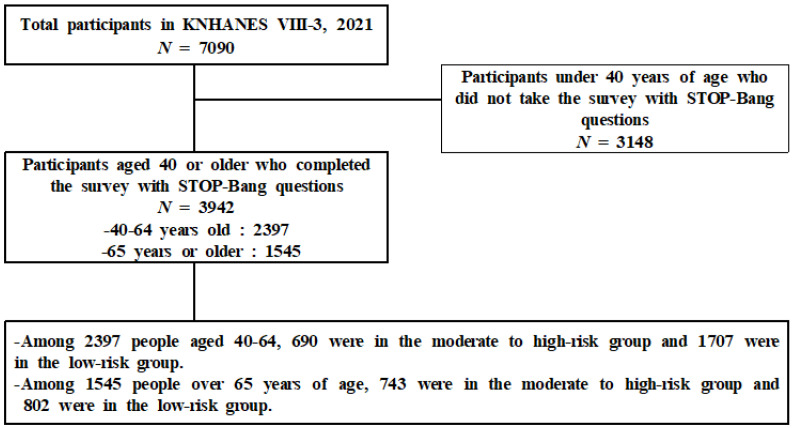
Participant selection process.

**Table 1 healthcare-11-03049-t001:** Comparison of factors related to obstructive sleep apnea according to age and OSA risk level.

Characteristics		Middle Age (*n* = 2397)			Elderly (*n* = 1545)		
		Low Risk (*n* = 1707)	Moderate–High Risk (*n* = 690)	x^2^ (*p*)	Low Risk (*n* = 802)	Moderate–High Risk (*n* = 743)	x^2^ (*p*)
		N (Weight %)			N (Weight %)		
Snoring	Yes	125 (6.8)	381 (58.5)	731.42 (<0.001)	14 (1.5)	203 (27.4)	250.91 (<0.001)
	No	1582 (93.2)	309 (41.5)		788 (98.5)	540 (72.6)	
Fatigue	Yes	351 (20.1)	335 (45.3)	152.31 (<0.001)	94 (11.5)	362 (47.9)	259.19 (<0.001)
	No	1356 (79.9)	355 (54.7)		708 (88.5)	381 (52.1)	
Observed apnea	Yes	26 (1.5)	205 (31.3)	449.05 (<0.001)	3 (0.3)	91 (12.9)	126.65 (<0.001)
	No	1681 (98.5)	485 (68.7)		799 (99.7)	652 (87.1)	
Hypertension	Yes	156 (9.0)	370 (49.9)	466.81 (<0.001)	282 (34.3)	582 (77.3)	299.45 (<0.001)
	No	1551 (91.0)	320 (50.1)		520 (65.7)	161 (22.7)	
Age (years)	≤50	826 (51.5)	155 (24.4)	156.85 (<0.001)	-	-	-
	>50	881 (48.5)	535 (75.6)		802 (100)	743 (100)	
BMI (kg/m^2^)	≤35	1704 (99.8)	678 (98.3)	14.53 (0.004)	802 (100)	732 (98.4)	18.04 (0.001)
	>35	3 (0.2)	12 (1.7)		-	11 (1.6)	
Neck circumference (cm)	≤41	1695 (99.1)	584 (82.8)	229.06 (<0.001)	802 (100)	708 (95.1)	54.06 (<0.001)
	>41	12 (0.9)	106 (17.2)		-	35 (4.9)	
Sex	Male	522 (37.3)	519 (80.4)	394.50 (<0.001)	174 (22.6)	502 (68.7)	344.63 (<0.001)
	Female	1185 (62.7)	171 (19.6)		628 (77.4)	241 (31.3)	

**Table 2 healthcare-11-03049-t002:** Comparison of general characteristics according to age and OSA risk level.

Characteristics		Middle Age (*n* = 2397)			Elderly (*n* = 1545)		
		Low Risk (*n* = 1707)	Moderate–High Risk (*n* = 690)	x^2^/t (*p*)	Low Risk (*n* = 802)	Moderate–High Risk (*n* = 743)	x^2^/t (*p*)
		N (Weight %)/m (SE)			N (Weight %)/m (SE)		
Household income level	Upper	631 (37.1)	251 (38.3)	3.32 (0.230)	84 (12.5)	78 (12.5)	0.58 (0.787)
	Middle	936 (55.8)	363 (52.8)		352 (45.8)	335 (47.7)	
	Lower	133 (7.1)	74 (8.9)		360 (41.6)	327 (39.8)	
Education level	≤Middle school	237 (11.5)	148 (18.1)	18.69 (<0.001)	585 (70.5)	499 (65.4)	5.64 (0.121)
	High school	693 (42.2)	272 (40.6)		145 (19.2)	145 (20.9)	
	≥College	775 (46.3)	270 (41.2)		70 (10.3)	99 (13.7)	
Living with spouse	Yes	1369 (88.8)	554 (89.1)	0.03 (0.869)	498 (63.9)	544 (74.8)	21.65 (<0.001)
	No	208 (11.2)	86 (10.9)		292 (36.1)	190 (25.2)	
Number of household members	1	167 (8.6)	81 (10.1)	15.49 (0.002)	231 (25.6)	170 (20.8)	11.75 (0.024)
	2	455 (23.8)	81 (30.6)		417 (48.8)	442 (57.4)	
	≥3	1085 (67.7)	371 (59.3)		154 (25.6)	131 (21.8)	
Occupation type	Professional, office worker	483 (28.0)	186 (29.4)	38.77 (<0.001)	23 (3.4)	41 (5.8)	29.13 (<0.001)
	Service, sales worker	283 (16.4)	91 (12.0)		63 (8.3)	26 (3.4)	
	Technician, agriculture, simple laborer	397 (24.5)	237 (35.4)		227 (26.6)	263 (33.9)	
	Unemployed	543 (31.2)	176 (23.2)		489 (61.7)	413 (56.9)	
Work type	Day work	1133 (87.0)	486 (86.2)	0.68 (0.742)	320 (89.1)	331 (89.2)	4.67 (0.210)
	Night work	103 (8.0)	46 (7.6)		21 (6.4)	15 (3.7)	
	Shift work	60 (5.0)	30 (5.9)		13 (4.6)	22 (7.1)	
Work hours (hours/week)		39.01 (0.45)	42.18 (0.76)	−3.60 (<0.001)	26.60 (1.30)	31.40 (1.24)	−2.85 (0.005)
Weight change	Weight loss	175 (10.4)	102 (14.0)	7.42 (0.084)	127 (15.3)	139 (16.8)	1.59 (0.542)
	Weight gain	461 (26.9)	175 (23.9)		104 (12.8)	10 (14.2)	
	No change	1070 (62.7)	413 (62.2)		566 (71.9)	499 (69.0)	
Binge drinking frequency	None	614 (48.6)	146 (26.3)	92.59 (<0.001)	222 (71.8)	183 (46.0)	62.23 (<0.001)
	About 1/month	322 (28.3)	165 (32.1)		62 (20.3)	107 (26.8)	
	About 1/week	194 (17.2)	150 (29.3)		20 (6.7)	66 (19.9)	
	Almost every day	62 (5.9)	64 (12.3)		5 (1.1)	27 (7.3)	
Smoking	Yes	552 (36.0)	454 (68.9)	221.10 (<0.001)	173 (22.9)	413 (58.9)	212.22 (<0.001)
	No	1154 (64.0)	236 (31.1)		628(77.1)	327 (41.1)	

M: mean, SE: standard error.

**Table 3 healthcare-11-03049-t003:** Comparison of physical and psychological factors according to age and OSA risk level.

Characteristics		Middle Age (*n* = 2397)			Elderly (*n* = 1545)		
		Low Risk (*n* = 1707)	Moderate–High Risk (*n* = 690)	x^2^/t (*p*)	Low Risk (*n* = 802)	Moderate–High Risk (*n* = 743)	x^2^/t (*p*)
		N (Weight %)/m (SE)			N (Weight %)/m (SE)		
Diabetes	Yes	108 (6.7)	129 (18.0)	64.67 (<0.001)	166 (20.9)	195 (26.6)	6.98 (0.024)
	No	1599 (93.3)	561 (82.0)		636 (79.1)	548 (73.4)	
MI	Yes	4 (0.3)	13 (2.0)	17.02 (0.001)	16 (2.5)	42 (5.9)	11.36 (0.014)
	No	1703 (99.7)	677 (98.0)		786 (97.5)	701 (94.1)	
Stroke	Yes	16 (1.0)	17 (2.5)	7.26 (0.016)	31 (3.4)	57 (7.8)	13.85 (0.001)
	No	1691 (99.0)	673 (94.5)		771 (96.6)	686 (92.2)	
Aerobic physical activity	Yes	719 (43.2)	284 (42.3)	0.18 (0.679)	222 (30.2)	205 (28.3)	0.65 (0.497)
	No	984 (56.8)	406 (57.7)		575 (69.8)	535 (71.7)	
Sitting time (hours/day)		8.53 (0.19)	8.79 (0.21)	−1.11 (0.267)	12.82 (0.72)	13.35 (0.73)	−0.51 (0.602)
Subjective health	Healthy	586 (34.3)	191 (28.7)	54.06 (<0.001)	226 (30.1)	186 (24.7)	6.23 (0.104)
	Moderate	877 (52.0)	303 (44.8)		364 (44.3)	333 (46.0)	
	Unhealthy	244 (13.7)	196 (26.5)		212 (25.6)	224 (29.2)	
Subjective body image	Thin	190 (11.9)	57 (8.7)	34.55 (<0.001)	164 (20.1)	134 (17.4)	4.18 (0.224)
	Moderate	680 (39.9)	211 (30.0)		344 (42.9)	307 (40.8)	
	Obese	836 (48.2)	422 (61.3)		293 (37.1)	301 (41.9)	
Stress	Feel less	1293 (76.6)	489 (71.6)	6.62 (0.031)	665 (84.4)	616 (82.5)	1.0 (0.320)
	Feeling a lot	413 (23.4)	201 (28.4)		134 (15.6)	125 (17.5)	
Depression	Yes	164 (9.0)	92 (10.8)	1.98 (0.185)	86 (10.9)	102 (13.8)	3.09 (0.179)
	No	1542 (91.0)	598 (89.2)		713 (89.1)	640 (86.2)	
Suicidal thoughts	Yes	50 (2.4)	40 (4.9)	9.78 (0.003)	33 (3.5)	32 (3.7)	0.02 (0.885)
	No	1656 (97.6)	650 (95.1)		768 (96.5)	709 (96.3)	
Anxiety		1.87 (0.09)	1.85 (0.13)	0.13 (0.891)	1.39 (0.12)	1.61 (0.16)	−1.24 (0.213)

MI: myocardial infarction, M: mean, SE: standard error.

**Table 4 healthcare-11-03049-t004:** Risk of OSA-related factors according to age.

Characteristics		Middle Age (*n* = 2397)		Elderly (*n* = 1545)	
		OR	95% CI	OR	95% CI
Education level (≥college)	≤Middle school	1.45	0.84–2.49	1.64	0.74–3.64
	High school	1.16	0.83–1.62	1.14	0.48–2.72
Living with spouse (no)	Yes	1.77	0.97–3.24	2.12	0.82–5.47
Number of household members (≥3)	1	1.52	0.75–3.04	1.05	0.30–3.58
	2	1.57	1.09–2.27	1.29	0.63–2.63
Occupation type (unemployed)	Professional, office worker	0.97	0.55–1.70	1.36	0.41–4.45
	Service, sales worker	0.74	0.38–1.42	0.59	0.21–1.63
	Technician, agriculture, simple laborer	0.97	0.56–1.67	0.95	0.39–2.34
Work hours		1.01	0.99–1.01	0.99	0.97–1.01
Binge drinking frequency (almost every day)	None	0.56	0.30–1.05	0.24	0.05–1.12
	About 1/month	0.79	0.45–1.40	0.38	0.08–1.69
	About 1/week	1.02	0.59–1.75	0.56	0.10–2.97
Smoking (yes)	No	0.30	0.27–0.69	0.23	0.12–0.44
Diabetes (yes)	No	0.43	0.53–1.69	0.54	0.26–1.10
MI (yes)	No	0.09	0.01–2.81	0.37	0.12–1.17
Stroke (yes)	No	1.93	0.35–10.47	0.30	0.07–1.20
Sitting time (hours/day)		1.01	0.98–1.04	1.01	0.98–1.04
Subjective health (unhealthy)	Healthy	0.64	0.40–1.03	0.30	0.12–0.78
	Moderate	0.60	0.39–0.90	0.43	0.17–1.10
Subjective body image (obese)	Thin	0.59	0.35–1.01	0.75	0.30–1.89
	Moderate	0.55	0.40–0.76	0.98	0.50–1.91
Stress (feel less)	Feeling a lot	0.78	0.55–1.12	1.68	0.78–3.60
Suicidal thoughts (yes)	No	0.91	0.35–2.34	1.75	0.41–7.45

MI: myocardial infarction, CI: confidence interval, OR: odds ratio.

## Data Availability

The KNHANES database is a publicly available, open-access repository (https://knhanes.kdca.go.kr/knhanes; accessed on 1 July 2023).

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
