# Peer review of "Factors Related to Obstructive Sleep Apnea According to Age: A Descriptive Study"

_healthcare, 2023, doi:10.3390/healthcare11233049_

Round 1

Reviewer 1 Report

Comments and Suggestions for Authors

Almost all parts of this paper (Abstract, Methods, Results, Discussion) require extensive revision.

Due to numerous flaws, in order to be able to perform the review correctly, it is necessary to solve the certain problems beforehand.

Some questions that require clarification:

  • Line 3, Line 83, and Lines 248-249: Define the study design (whether it was `descriptive` or `cross-sectional` study design) that was applied in this study and harmonize it throughout the paper.
  • Lines 17-18: Check if the variables are specified correctly. Match this with the information in the Results section.
  • Line 87: Add a new paragraph with detailed information on `Study sample`, `Study sample size calculation`, with criteria for inclusion and with criteria for exclusion from the study, `Partitipation rate`, `Response rate`.
  • Precisely define (specify age) which persons were in the `middle-aged` group and which were in the `elderly` group.
  • Lines 87-89: This text presents the results of this work. Move this paragraph in the Results section.
  • Lines 168-169: Is this description of the data from Table 4 correct? Check and correct.
  • Lines 192-239: The data listed in the Results section showed which variables were statistically significantly associated with obstructive sleep apnea (by age). Unfortunately, the text in the Discussion section does not take into account the presented own results, but each observed variable is discussed (very wrongly) as if each of them was significantly associated with obstructive sleep apnea (by age). Correct the complete text of the Discussion section in such a way as to accurately comment on one's own results, in comparison with the results of similar studies by other authors / in other countries.

Comments on the Quality of English Language

The quality of English language is appropriate.  

Author Response

Thank you for your careful review. We revised the paper to reflect the reviewer's comments. Modifications are highlighted in red in the text. Thank you for your reasonable and thoughtful review.

Reviewer 1.

Almost all parts of this paper (Abstract, Methods, Results, Discussion) require extensive revision.

Due to numerous flaws, in order to be able to perform the review correctly, it is necessary to solve the certain problems beforehand.

Some questions that require clarification:

  • Line 3, Line 83, and Lines 248-249: Define the study design (whether it was `descriptive` or `cross-sectional` study design) that was applied in this study and harmonize it throughout the paper.

Answer: After discussion, we decided to use the word descriptive study.

  • Lines 17-18: Check if the variables are specified correctly. Match this with the information in the Results section.

Answer: Matched the information with the Results section.

  • Line 87: Add a new paragraph with detailed information on `Study sample`, `Study sample size calculation`, with criteria for inclusion and with criteria for exclusion from the study, `Partitipation rate`, `Response rate`.

Answer: This study is a secondary analysis of data from the Eighth Korea National Health and Nutrition Examination Survey (KNHANES VIII-3). KNHANES is a statutory survey on citizens' health behavior and the status of chronic diseases based on Article 16 of the National Health Promotion Act of the Republic of Korea. This study was conducted with data obtained with permission from the Korea Disease Control and Prevention Agency for academic purposes. And according to the recommendations of the Korea Disease Control and Prevention Agency, the analysis was conducted using complex sample analysis.

Participants were described in Study Design and Participants as follows.

‘In total, 7,090 people were included in the KNHANES VIII-3 (2021). Of these, 3,942 participated in this study: 2,397 middle-aged (40-64 years old) and 1,545 elderly (65 years or older) individuals. Among 2,397 middle-aged people at risk for OSA, 690 were at moderate-to-high risk and 1,707 were at low risk. Among 1,545 older adults at risk of OSA, 743 were at moderate-to-high risk and 802 were at low risk.’

Figure 1. Participant selection process

(Figure is in PDF file)

  • Precisely define (specify age) which persons were in the `middle-aged` group and which were in the `elderly` group.

Answer: Age was added to the sentence as shown below.

‘In total, 7,090 people were included in the KNHANES VIII-3 (2021). Of these, 3,942 participated in this study: 2,397 middle-aged (40-64 years old) and 1,545 elderly (65 years or older) individuals.

  • Lines 87-89: This text presents the results of this work. Move this paragraph in the Results section.

Answer: Part of the content has been moved to the Results section.

  • Lines 168-169: Is this description of the data from Table 4 correct? Check and correct.

Answer: The contents have been modified as follows.

‘Factors associated with OSA in middle-aged individuals included number of household members, smoking, subjective health, and subjective body image.’

  • Lines 192-239: The data listed in the Results section showed which variables were statistically significantly associated with obstructive sleep apnea (by age). Unfortunately, the text in the Discussion section does not take into account the presented own results, but each observed variable is discussed (very wrongly) as if each of them was significantly associated with obstructive sleep apnea (by age). Correct the complete text of the Discussion section in such a way as to accurately comment on one's own results, in comparison with the results of similar studies by other authors / in other countries.

Answer: This study aimed to identify the factors associated with OSA by age, and for this purpose, a secondary analysis was performed on KNHANES VIII-3 (2021) data. As a result, it was reported that OSA related factors differ depending on the age (middle-aged and elderly) of OSA patients. The discussion section covers related characteristics according to OSA risk, factors according to age and OSA risk, and OSA-related factors according to age. It also mentioned areas where the results were different from what we expected.

While revising the discussion, we referred to papers similar to this study (*D. W. Lee & J. Lee, 2022; C. Stepnowsky et al., 2019), and ask you to consider the points mentioned above.

*Lee, D. W.; Lee, J. The association between long working hours and obstructive sleep apnea assessed by STOP-BANG score: a cross-sectional study. Int Arch Occup Environ Health 2023, 96, 191–200. DOI:10.1007/s00420-022-01914-z.

*Stepnowsky, C.; Sarmiento, K. F.; Bujanover, S.; Villa, K. F.; Li, V. W.; Flores, N. M. Comorbidities, Health-Related Quality of Life, and Work Productivity Among People With Obstructive Sleep Apnea With Excessive Sleepiness: Findings From the 2016 US National Health and Wellness Survey. J Clin Sleep Med 2019, 15, 235-243. DOI: 10.5664/jcsm.7624

Reviewer 2 Report

Comments and Suggestions for Authors

The manuscript by Kwon and colleagues evaluates the factors associated with obstructive sleep apnea in middle-aged and elderly people separately. Overall the manuscript is well-written and this is an interesting study. I have a few comments that I think could help strengthen the presentation of the methods and results.

  • You mentioned that 7,090 people were included in the KNHANES VIII-3, 2021. Of these, 3,942 participated in this study. How these 3,942 were selected in this study? Could you provide the inclusion criteria? How the rest of the individuals were excluded? Did these 3,942 respond to all the questions from the questionnaire? Do you have data regarding the physical and psychological characteristics of those (7,090-3,942) people who did not participate in this study? Did you see any patterns? Are there significant differences in the physical and psychological characteristics between the study participants vs. the participants who were excluded from the study? Significant differences could introduce selection bias or generalizability issues to this study. It would be great to mention these in the discussion.

  • In section 2.4 statistical analyses, is the p-value 2-sided or 1-sided?

  • Table 4 is a bit confusing. Does it show the results from a multivariate logistic regression with all the independent characteristics included in the same model? Or does it show the results from several univariate logistic regression with the independent characteristics in the separate models? If that’s the former, can you provide R squared, AIC, BIC, -2loglikelihood, or conduct a goodness of fit test to demonstrate the model fit? If that’s the latter, could you fit a multivariate logistic regression with a proper variable selection process?

Author Response

Thank you for your careful review. We revised the paper to reflect the reviewer's comments. Modifications are highlighted in red in the text. Thank you for your reasonable and thoughtful review.

Reviewer 2.

The manuscript by Kwon and colleagues evaluates the factors associated with obstructive sleep apnea in middle-aged and elderly people separately. Overall the manuscript is well-written and this is an interesting study. I have a few comments that I think could help strengthen the presentation of the methods and results.

  • You mentioned that 7,090 people were included in the KNHANES VIII-3, 2021. Of these, 3,942 participated in this study. How these 3,942 were selected in this study? Could you provide the inclusion criteria? How the rest of the individuals were excluded? Did these 3,942 respond to all the questions from the questionnaire? Do you have data regarding the physical and psychological characteristics of those (7,090-3,942) people who did not participate in this study? Did you see any patterns? Are there significant differences in the physical and psychological characteristics between the study participants vs. the participants who were excluded from the study? Significant differences could introduce selection bias or generalizability issues to this study. It would be great to mention these in the discussion.

Answer: This study is a secondary analysis of data from the Eighth Korea National Health and Nutrition Examination Survey (KNHANES VIII-3). KNHANES is a statutory survey on citizens' health behavior and the status of chronic diseases based on Article 16 of the National Health Promotion Act of the Republic of Korea. This study was conducted with data obtained with permission from the Korea Disease Control and Prevention Agency for academic purposes. And according to the recommendations of the Korea Disease Control and Prevention Agency, the analysis was conducted using complex sample analysis.

The Korea Disease Control and Prevention Agency recommends that weights be assigned to correct for uneven extraction rates and non-response errors during analysis. This study was also analyzed by assigning weights to the health survey according to the recommendations of the Korea Disease Control and Prevention Agency. When weighted complex sample analysis is performed, the representativeness and accuracy of estimates can be improved. (from: Guidelines for using raw data from the National Health and Nutrition Examination Survey, 2023, Korea Disease Control and Prevention Agency)

Please refer to Figure 1 for the participant selection process.

We added weight to the discussion section to reduce confusion.

  • In section 2.4 statistical analyses, is the p-value 2-sided or 1-sided?

Answer: It is 2-sided.

  • Table 4 is a bit confusing. Does it show the results from a multivariate logistic regression with all the independent characteristics included in the same model? Or does it show the results from several univariate logistic regression with the independent characteristics in the separate models? If that’s the former, can you provide R squared, AIC, BIC, -2loglikelihood, or conduct a goodness of fit test to demonstrate the model fit? If that’s the latter, could you fit a multivariate logistic regression with a proper variable selection process?

Answer: This study performed complex sample logistic analysis after assigning weights according to the recommendations of the Korea Disease Control and Prevention Agency. Unlike general logistic analysis, complex sample logistic analysis does not present many results.

Characteristics that were significantly different depending on the severity of obstructive sleep apnea were analyzed by entering them as independent variables in a complex sample logistic analysis.

Reviewer 3 Report

Comments and Suggestions for Authors

COMMENTS TO THE AUTHORS

Thank you for inviting me to review the manuscript entitled “Factors related to Obstructive Sleep Apnea According to Age: a Descriptive Study”.

This is a secondary analysis of the findings derived from Eight Korea National Health and Nutrition Examination Survey, 2021. It is based on a total of 3942 individuals and aims at identifying significantly different variables between middle-age and elderly individuals with low or high risk of OSA.

There are many major concerns to this study. My major concern is the arbitrary cut-off of 50 to distinguish between middle -age and elderly, cut-off that is at the basis of the entire study and findings. More details need to be provided on the original study KNHANES and on the selection of the participants. The STOP-Bang has been classified utilizing a binary system, when originally this screening tool provides a low, moderate and high-risk classification. This has not been supported by any reference in this study. There is also lack of description of the methodology: how were the variables assessed? STOP-Bang was also utilized to assess OSA severity, which is not accurate, as this is only a screening tool that cannot diagnose OSA. More details to my concerns are described below. 

Abstract: If word permits, it would be valuable adding how the groups were classified in low or high risk (line 15). 

Line 19: in the lines prior, the “older adult” group was defined as “elderly”. Authors should be consistent with the terminology

Introduction:

- Line 35: if “prevalence of OSA increases with age”, the highest prevalence should be seen in the elderly. I suggest that the authors rephrase the sentence, not to sound as contradictory. 

- Line 38: here it seems like aerobic physical activity was related to the occurrence and severity of OSA. I suggest that the authors rephrase the concept, such as “decreased aerobic physical activity”.

- Reference 8: this needs to be changed, as this reference does not mention that 80% of individuals with OSA are undiagnosed. 

- Line 55: it may be valuable to mention which “important diseases” smoking is connected to

- I am in doubt whether reference 15 actually talks about increased drug compliance in patients with depression, but I don’t have access to the full text of the article

- line 74-75: the authors need to clarify to a non-expert reader that STOP-Bang contains an item that assess age of 50, so that the cut-off to distinguish between middle-age and elderly adults was chosen as 50. However, this cut-off is arbitrary. Did the author confirm that when the literature distinguishes between middle-age and elderly utilize 50 years old as cut-off? I agree that this was a valuable tool to distinguish between two age categories. However, the findings of this study are not consistent with middle-age vs elderly (that normally in OSA is intended to be way older than 50 years old), but only a distinction between < 50 and > 50 years old. This need to be accurately highlighted among the limitations of the study. 

 Methods:

- It is not clear how the 3,942 individuals who participated in this study were selected. Was this a selection done by the authors? Based on age? Based on what? 

- It is not clear if these participants had OSA, and what the KNHANES VIII study investigated originally, whether it was a nationally-based study, who the participants were, etc. This is important information to understand the selection of the participants. 

- It is not clear how the participants were classified at low risk or at high risk (line 88-89)

- “OSA was measured with STOP-Bang” (line 91) is not correct. STOP-Bang is a screening tool. Do the authors intended that STOP-bang was used to classify the participants in low, moderate or high risk of having OSA? Moreover, the authors need to provide reference that support why the STOP-Bang was scored and read in that way. Normally, the values norms are the following: 1-2 is low-risk, 3-4 is moderate, >5 is high. This different classification is at the basis of the entire study. I do not believe is accurate nor supported by the most solid literature, but at least the authors need to provide references and studies that supported this different classification.

- paragraph lines 97-114: the authors need to explicitly stated how these variables were assessed. This still derives from the lack of details on the original study. Did this derive from an interview to the participants? Was this self-reported? Was this extracted from electronic medical records? All these details need to be provided. 

- How were depression and suicidal thoughts assessed? Was there a temporal relationship indicated in the question? Were these ad hoc questions or validated questionnaires? 

- statistical analyses: this paragraph should reflect not only which statistical test has been used, that here is correctly identified, but also how it was used. Was it used to identify the predictors of low vs high risk OSA? Was it used to distinguish between different age categories? Healthy style? Etc This info is not provided, so this passage is obscure 

Results: 

This section should start with a general description of the sample size, which is not provided here. There is no indication in the statistical analysis that low vs high risk groups will be compared based on single items of STOP-Bang. It is not even clear that this was done based on stop-bang. 

It is inappropriate to categorize OSA as “severity of OSA” (for example on line 157), because the STOP-Bang does not provide a diagnosis of OSA, but rather assess the high, moderate or low risk of having OSA. This is only a screening tool and not a diagnostic tool. It assess the risk, not the severity of disease. 

- Line 160-161: which group do these findings refer to? 

- section 3.4 Factors associated with OSA by age. This is not clear how “OSA” was classified. This goes back to the fact that is not clear from the methodology section if these participants are really diagnosed with OSA or not. Do the authors mean “high-risk of OSA” or were these participants accurately diagnosed with a PSG read by a sleep physician? 

Discussion: 

- Line 204-205: this seems more like a speculation. Contributing factors likely play a role here. What about depression and suicidal intention? What about physical habits contributing to this? This sort of speculation can only be done if every single variable is investigated by controlling for other covariates  

Author Response

Thank you for your careful review. We revised the paper to reflect the reviewer's comments. Modifications are highlighted in red in the text. Thank you for your reasonable and thoughtful review.

Reviewer 3.

Thank you for inviting me to review the manuscript entitled “Factors related to Obstructive Sleep Apnea According to Age: a Descriptive Study”.

This is a secondary analysis of the findings derived from Eight Korea National Health and Nutrition Examination Survey, 2021. It is based on a total of 3942 individuals and aims at identifying significantly different variables between middle-age and elderly individuals with low or high risk of OSA.

There are many major concerns to this study. My major concern is the arbitrary cut-off of 50 to distinguish between middle -age and elderly, cut-off that is at the basis of the entire study and findings. More details need to be provided on the original study KNHANES and on the selection of the participants. The STOP-Bang has been classified utilizing a binary system, when originally this screening tool provides a low, moderate and high-risk classification. This has not been supported by any reference in this study. There is also lack of description of the methodology: how were the variables assessed? STOP-Bang was also utilized to assess OSA severity, which is not accurate, as this is only a screening tool that cannot diagnose OSA. More details to my concerns are described below. 

Abstract: If word permits, it would be valuable adding how the groups were classified in low or high risk (line 15). 

Answer: How to distinguish between moderate to high-risk and low-risk groups for OSA is described in Study Variables. If this is entered into abstract, the abstract becomes too long. Please understand what is described in Study variables.

Line 19: in the lines prior, the “older adult” group was defined as “elderly”. Authors should be consistent with the terminology

Answer: Older adults changed to elderly.

Introduction:

- Line 35: if “prevalence of OSA increases with age”, the highest prevalence should be seen in the elderly. I suggest that the authors rephrase the sentence, not to sound as contradictory. 

Answer: The sentence was modified as follows.

The prevalence of OSA has been reported to differ by age, with an increase in those aged <60 years but a decrease in those aged >60 years, prompting the need to investigate differences by life cycle [4].

- Line 38: here it seems like aerobic physical activity was related to the occurrence and severity of OSA. I suggest that the authors rephrase the concept, such as “decreased aerobic physical activity”.

Answer: The following sentence was added to emphasize the relationship between physical activity and OSA risk factors.

Physical activity has been reported to lower OSA risk. Studies have shown that increasing physical activity to 15 MET-h per week reduces OSA risk by approximately 10% in both men and women [5].

- Reference 8: this needs to be changed, as this reference does not mention that 80% of individuals with OSA are undiagnosed. 

Answer: The text and references have been revised as follows.

It has been reported that 26% of adults suffer from mild to severe OSA, and this prevalence is

expected to increase further [8].

  1. Peppard, P.E.; Young, T.; Barnet, J.H.; Palta, M.; Hagen, E.W.; Hla, K.M. Increased prevalence of sleep-disordered breathing in adults. Am J Epidemiol 2013, 177, 1006–1014. DOI: 1093/aje/kws342

- Line 55: it may be valuable to mention which “important diseases” smoking is connected to

Answer: Modified to cardiovascular disease.

- I am in doubt whether reference 15 actually talks about increased drug compliance in patients with depression, but I don’t have access to the full text of the article

Answer: Reference number 15 is attached. The sentence was modified as follows.

A previous study has reported that the presence of OSA in patients with depression influences the treatment of depression and the effectiveness of medication [15].

- line 74-75: the authors need to clarify to a non-expert reader that STOP-Bang contains an item that assess age of 50, so that the cut-off to distinguish between middle-age and elderly adults was chosen as 50. However, this cut-off is arbitrary. Did the author confirm that when the literature distinguishes between middle-age and elderly utilize 50 years old as cut-off? I agree that this was a valuable tool to distinguish between two age categories. However, the findings of this study are not consistent with middle-age vs elderly (that normally in OSA is intended to be way older than 50 years old), but only a distinction between < 50 and > 50 years old. This need to be accurately highlighted among the limitations of the study. 

Answer: As you pointed out, in order to correct the age classification standards, we deleted incorrect sentences and modified the sentences in the introduction.

In this study, the participants were divided into middle-aged (aged 41 to 64) and elderly (aged 65 or older). Describe the participant's age classification in 2.1. This was done in Study Design and Participants.

The sentence was modified as follows.

 The STOP-Bang questionnaire is an easy to evaluate tool comprising eight questions [17] and has been reported to be a valid tool in a recent meta-analysis [18]. Therefore, this study aimed to identify factors influencing OSA among middle-aged and elderly people, as they are major groups that, to the best of our knowledge, have not been evaluated separately until now.

 Methods:

- It is not clear how the 3,942 individuals who participated in this study were selected. Was this a selection done by the authors? Based on age? Based on what? 

Answer: As shown in the figure below, excluding 3,148 participants under the age of 40 who did not take the survey, the number of eligible participants was 3,942. The eligible participants were then divided into middle-aged and elderly.

Figure 1. Participant selection process

(Figure is in PDF file)

- It is not clear if these participants had OSA, and what the KNHANES VIII study investigated originally, whether it was a nationally-based study, who the participants were, etc. This is important information to understand the selection of the participants. 

Answer: KNHANES is a statutory survey on citizens' health behavior and the status of chronic diseases based on Article 16 of the National Health Promotion Act of the Republic of Korea. This survey conducted by the Korea Disease Control and Prevention Agency is also used as basic data for health policies, such as the development of programs to promote public health. KNHANES is a country-based study, and the subjects are Korean citizens. In order to extract a representative sample, a two-stage stratified cluster sampling method was applied to citizens residing in Korea.

Hypertension, body mass index, age, neck circumference, and sex are variables that have previously been continuously measured, and snoring, fatigue, and observed apnea were added in 2021 to assess obstructive sleep apnea.

- It is not clear how the participants were classified at low risk or at high risk (line 88-89)

Answer: It is described in the Study variables section as follows.

If the participants answered “yes” to three or more questions, they were classified into the “moderate to high-risk group,” and if they answered “yes” to less than three questions, they were classified into the “low-risk group.”

- “OSA was measured with STOP-Bang” (line 91) is not correct. STOP-Bang is a screening tool. Do the authors intended that STOP-bang was used to classify the participants in low, moderate or high risk of having OSA? Moreover, the authors need to provide reference that support why the STOP-Bang was scored and read in that way. Normally, the values norms are the following: 1-2 is low-risk, 3-4 is moderate, >5 is high. This different classification is at the basis of the entire study. I do not believe is accurate nor supported by the most solid literature, but at least the authors need to provide references and studies that supported this different classification.

Answer: As recommended by the reviewer, high risk was changed to moderate-high risk.

- paragraph lines 97-114: the authors need to explicitly stated how these variables were assessed. This still derives from the lack of details on the original study. Did this derive from an interview to the participants? Was this self-reported? Was this extracted from electronic medical records? All these details need to be provided. 

Answer: The following information has been added to Study Design and Participants.

A trained investigator met with the participants in person and administered the survey using a self-report method.

- How were depression and suicidal thoughts assessed? Was there a temporal relationship indicated in the question? Were these ad hoc questions or validated questionnaires? 

Answer: The depression question asks whether you have been depressed to the extent that it interferes with your daily life for more than two consecutive weeks in the past year. The suicidal ideation question asks whether you have ever thought about suicide in the past year.

- statistical analyses: this paragraph should reflect not only which statistical test has been used, that here is correctly identified, but also how it was used. Was it used to identify the predictors of low vs high risk OSA? Was it used to distinguish between different age categories? Healthy style? Etc This info is not provided, so this passage is obscure 

Answer: The Statistical Analyzes section describes the following:

  1. The differences and degrees of variables according to the participants’ age and degree of OSA were analyzed using complex sample descriptive analysis and the chi-squared test.
  2. Factors related to OSA according to age were identified using complex sample logistic regression.

Results: 

This section should start with a general description of the sample size, which is not provided here. There is no indication in the statistical analysis that low vs high risk groups will be compared based on single items of STOP-Bang. It is not even clear that this was done based on stop-bang. 

It is inappropriate to categorize OSA as “severity of OSA” (for example on line 157), because the STOP-Bang does not provide a diagnosis of OSA, but rather assess the high, moderate or low risk of having OSA. This is only a screening tool and not a diagnostic tool. It assess the risk, not the severity of disease. 

Answer: STOP-Bang was explained in Study Variables, and comparison of the two groups was described in Statistical Analysis.

Some information about the participants has been added to the results section.

Severity of OSA has been modified to OSA risk level.

- Line 160-161: which group do these findings refer to? 

Answer: Line 161 is the results for two groups: middle-aged and elderly. Both groups showed differences in diabetes, myocardial infarction, and stroke depending on the risk of OSA.

- section 3.4 Factors associated with OSA by age. This is not clear how “OSA” was classified. This goes back to the fact that is not clear from the methodology section if these participants are really diagnosed with OSA or not. Do the authors mean “high-risk of OSA” or were these participants accurately diagnosed with a PSG read by a sleep physician? 

Answer: This means that the risk of OSA was measured using STOP-Bang, rather than being diagnosed by a sleep physician.

In this study, OSA was classified into low risk and moderate-high risk.

Discussion: 

- Line 204-205: this seems more like a speculation. Contributing factors likely play a role here. What about depression and suicidal intention? What about physical habits contributing to this? This sort of speculation can only be done if every single variable is investigated by controlling for other covariates  

Answer: As recommended by the reviewer, the contents have been modified as follows:

The results of this study showed that working hours were associated with the severity of OSA in both middle-aged and elderly. These results are similar to a paper evaluating KNHNES 2020 data using the STOP-BANG score, which stated that long working hours or overtime increases OSA risk [26]. Additionally, it was said that occupations that require minimal physical activity due to sitting for long periods may contribute to OSA due to lack of physical activity and obesity [27, 28]. Nevertheless, OSA may be influenced by other mediating factors other than age and working hours, so further investigation is needed.

Round 2

Reviewer 1 Report

Comments and Suggestions for Authors

In the revised version of this paper, the authors made very important corrections, which contributed to the clarity and transparency of the presented results. The authors responded to my comments, answered some questions correctly and provided satisfactory explanations. Thanks to the authors. 

Comments on the Quality of English Language

The quality of English language is appropriate. 

Author Response

We appreciate your careful review.

Thank you.

Reviewer 3 Report

Comments and Suggestions for Authors

The authors have now provided a satisfactory response. I still want to point out some minor corrections that need to be amended. 

Introduction:

line 74-77. The STOP-bang needs further detail. It is a tool for what? IN addition, sentence in line 75 is messed up. 

Methods: 

Line 97 "OSA was measured using the STOP-Bang tool" is not correct, as already pointed out in my previous revision. The STOP-Bang does not measure OSA, but rather may quantify the "risk of OSA". Please, modify this to reflect a more appropriate statement on STOP-Bang. 

Legend of Table 4 also needs to reflect that we are talking about risk of OSA, not OSA. 

Author Response

Reviewer 3. (Round 2)

Thank you for your careful consideration and advice.

Introduction:

  • line 74-77. The STOP-bang needs further detail. It is a tool for what? IN addition, sentence in line 75 is messed up.

Answer: It was stated that STOP-Bang is a tool to evaluate the risk of OSA. We also modified sentences that needed correction.

Additionally, it is essential to assess the risk of obstructive sleep apnea (OSA) and identify differences. A convenient and reliable tool is necessary to evaluate the risk of OSA. As a tool for assessing the risk of OSA, STOP-Bang comprises 8 questions, providing a convenient means of evaluation [17]. Furthermore, a recent meta-analysis study [18] has validated its effectiveness, confirming its reliability as an assessment tool.

Methods:

  • Line 97 "OSA was measured using the STOP-Bang tool" is not correct, as already pointed out in my previous revision. The STOP-Bang does not measure OSA, but rather may quantify the "risk of OSA". Please, modify this to reflect a more appropriate statement on STOP-Bang.

Answer: The sentence regarding the content you pointed out has been modified to read 'Risk of OSA was measured using the STOP-Bang tool'.

  • Legend of Table 4 also needs to reflect that we are talking about risk of OSA, not OSA.

Answer: The title of Result 3.4 has been revised to ‘Factors Associated with risk of OSA by Age’. And the title of Table 4 was changed to ‘Risk of OSA-related factors according to age’. Some of the contents of Result 3.4 were modified from OSA to ‘risk of OSA’.
